# A randomized trial in a massive online open course shows people don't know what a statistically significant relationship looks like, but they can learn

Aaron Fisher[1], G. Brooke Anderson[2], Roger Peng[1] and Jeff Leek[1]

[1] Department of Biostatistics, Johns Hopkins Bloomberg School of Public Health, Baltimore, MD, USA
[2] Department of Environmental & Radiological Health Sciences, Colorado State University, Fort Collins, CO, USA

## ABSTRACT

Scatterplots are the most common way for statisticians, scientists, and the public to visually detect relationships between measured variables. At the same time, and despite widely publicized controversy, *P*-values remain the most commonly used measure to statistically justify relationships identified between variables. Here we measure the ability to detect statistically significant relationships from scatterplots in a randomized trial of 2,039 students in a statistics massive open online course (MOOC). Each subject was shown a random set of scatterplots and asked to visually determine if the underlying relationships were statistically significant at the $P < 0.05$ level. Subjects correctly classified only 47.4% (95% CI [45.1%–49.7%]) of statistically significant relationships, and 74.6% (95% CI [72.5%–76.6%]) of non-significant relationships. Adding visual aids such as a best fit line or scatterplot smooth increased the probability a relationship was called significant, regardless of whether the relationship was actually significant. Classification of statistically significant relationships improved on repeat attempts of the survey, although classification of non-significant relationships did not. Our results suggest: (1) that evidence-based data analysis can be used to identify weaknesses in theoretical procedures in the hands of average users, (2) data analysts can be trained to improve detection of statistically significant results with practice, but (3) data analysts have incorrect intuition about what statistically significant relationships look like, particularly for small effects. We have built a web tool for people to compare scatterplots with their corresponding *p*-values which is available here: http://glimmer.rstudio.com/afisher/EDA/.

# INTRODUCTION

Over the last two decades there has been a dramatic increase in the amount and variety of data available to scientists, physicians, and business leaders in nearly every area of application. Statistical literacy is now critical for anyone consuming data analysis reports,

Corresponding author
Jeff Leek, jleek@jhsph.edu

including scientific papers, newspaper reports (*Beyth-Marom, Fidler & Cumming, 2008*), legal cases (*Gastwirth, 1988*), and medical test results (*Schwartz et al., 1997*; *Sheridan, Pignone & Lewis, 2003*). A lack of sufficient training in statistics and data analysis has been responsible for the retraction of high-profile papers (*Ledford, 2011*), the cancellation of clinical trials (*Pelley, 2012*), and mistakes in papers used to justify major economic policy initiatives (*Cassidy, 2013*).

Despite the critical importance of statistics and data analysis in modern life, we have relatively little empirical evidence about how statistical tools work in the hands of typical analysts and consumers. The most well-studied statistical tool is the visual display of quantitative information. Previous studies have shown that humans have difficulty interpreting linear measures of correlation (*Cleveland, 1982*), are better at judging relative positions than relative angles (*Heer & Bostock, 2010*; *Cleveland & McGill, 1985*), and view correlations differently when plotted on different scales (*Cleveland, 1982*). These studies show that mathematically correct statistical procedures may have unintended consequences in the hands of users. The real effect of a statistical procedure depends, to a large extent, on psychology and cognitive function.

Here we perform a large-scale study of the ability of average data analysts to detect statistically significant relationships from scatterplots. Our study compares two of the most common data analysis tasks, making scatterplots and calculating *P*-values. It has been estimated that as many as 80% of the plots published across all scientific disciplines are scatterplots (*Tufte & Graves-Morris, 1983*). At the same time, and despite widely publicized controversy over their use (*Nuzzo, 2014*), *P*-values remain the most common choice for reporting a statistical summary of the relationship between two variables in the scientific literature. In the decade 2000–2010, 15,653 *P*-values were reported in the abstracts of the *The Lancet*, *The Journal of the American Medical Association*, *The New England Journal of Medicine*, *The British Medical Journal*, and *The American Journal of Epidemiology* (*Jager & Leek, 2007*).

Data analysts frequently use exploratory scatterplots for model selection and building. Selecting which variables to include in a model can be viewed as visual hypothesis testing where the test statistic is the plot and the measure of significance is human judgement. However, it is not well known how accurately humans can visually classify significance when looking at graphs of raw data. This classification task depends on both understanding what combinations of sample size and effect size constitute significant relationships, and being able to visually distinguish these effect sizes. We performed a set of experiments to (1) estimate the baseline accuracy with which subjects could visually determine if two variables showed a statistically significant relationship; (2) test whether accuracy in visually classifying significance was changed by the number of data points in the plot or the way the plot was presented; and (3) test whether accuracy in visually classifying significance improved with practice. Our intuition is that potential improvements with practice would be better explained by an improved cognitive understanding of statistical significance, rather than an improved perceptive ability to distinguish effect sizes.

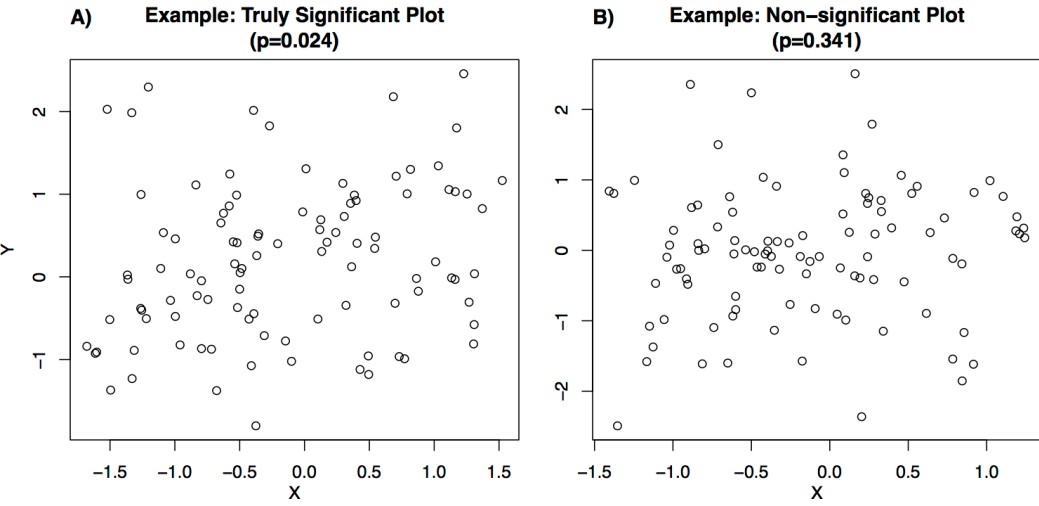

**Figure 1** Examples of plots shown to users.

## METHODS AND RESULTS

Our study was conducted within the infrastructure of a statistics massive online open course (MOOC). While MOOCs have previously been used to study MOOCs (*Do et al., 2013*; *Mak, Williams & Mackness, 2010*; *Liyanagunawardena, Adams & Williams, 2013*), to our knowledge this is the first example of a MOOC being used to study the practice of science. Specifically, our survey was conducted as an ungraded, voluntary exercise within the Spring 2013 Data Analysis Coursera class. This class was 8 weeks long, and consisted of lecture content, readings, and a weekly quiz. Although 121,257 students registered for the course, only 5,306 completed the final weekly quiz. The survey was made available to all students in the class, and 2,039 students responded—approximately 38% relative to the number of active users. In one of the weekly quizzes preceding the survey, students were asked two questions relating to the concept of *P*-values (see Supplemental Information for specific question text). Students had two attempts at each question and their accuracies were: 73.5% (1st attempt, 1st question), 96.1% (2nd attempt, 1st question), 73.1% (1st attempt, 2nd question), and 95.4% (2nd attempt, 2nd question). These questions were not identical to the questions in our survey but suggest that students understand the concepts behind a *P*-value, assuming that almost all students completed the graded quizzes before submitting responses to the optional exercises that followed.

Each student who participated in the survey was shown a set of bi-variate scatterplots (examples shown in Fig. 1). The set of plots included eight plots from seven different categories (Table 1), with two plots from the reference category (of which one was significant and one was not) and one plot from each of the other categories (each randomly chosen to be either significant or non-significant). These plot categories (Table 1) were selected to allow analysis of whether students' accuracy in visually classifying significance changed based on the number of data points in the plot, or the plot's presentation style. Each set of plots shown to a user was randomly selected from a library containing 10 plots from each category (see Supplemental Information for full library and generating code),

| Table 1 Plot categories shown to users. | |
|---|---|
| Reference | 100 data points (e.g., Fig. 1) |
| Smaller $n$ | 35 data points |
| Larger $n$ | 200 data points |
| Best-fit line | 100 data points, with best fit line added |
| Lowess | 100 data points, with smooth lowess curve added (using R "lowess" function) |
| Axis Scale | 100 data points, with the axis range increased to 1.5 standard deviations outside $X$ and $Y$ variable ranges (e.g., "zoomed out") (*Cleveland, 1982*) |
| Axis Label | 100 data points, with fictional $X$- and $Y$-axis labels added corresponding to activation in a brain region (e.g., "Cranial Electrode 33 (Standardized)" versus "Cranial Electrode 92 (Standardized)") |

of which half were statistically significant (*P*-values from testing the slope coefficient in a linear regression relating $X$ and $Y$ were between 0.023 and 0.025; e.g., Fig. 1A) and half were not statistically significant (*P*-values between 0.33 and 0.35; e.g., Fig. 1B).

For each plot, students were asked to visually determine whether the bi-variate relationship shown was statistically significant at the 0.05 level (in the example plots shown in Fig. 1, the correct answer would have been "statistically significant" for Fig. 1A for which the *P*-value of a linear relationship between the $X$ and $Y$ variables is 0.024, and "not statistically significant" for Fig. 1B for which the *P*-value is 0.341). All eight plots were shown at the same time and students submitted responses for all plots in a single submission. Students were also able to submit a partial response by leaving some of the survey questions blank. 94.4% of users completed their first attempt of the survey. After submitting their responses, students were shown the correct answers and given the opportunity to retake the survey with a new set of plots. Students were not told any information about the structure of the survey (e.g., the fact that one of the "Reference" plots was significant and one was not) to improve their accuracy.

To analyze responses, we created separate models for the probability of correctly visually classifying significance in: (1) graphs that showed two variables with a statistically significant relationship (e.g., Fig. 1A) and (2) graphs that showed two variables with a statistically non-significant relationship (e.g., Fig. 1B). These two types of visual classification correspond to the separate accuracy metrics: human sensitivity to significance (accuracy in giving a positive result in cases where a condition is true) and human specificity to non-significance (accuracy in giving a negative result in cases where a condition is false). In this framework, the hypothetical baseline case where humans have no ability to classify significance corresponds to the sensitivity rate being equal to one minus the specificity rate, which means that the probability of visually classifying a plot as significant is unaffected by the actual significance level of the plot. Accuracy in both metrics was modeled by logistic regressions with person-specific random intercept terms, using the "lme4" package in R (see Supplemental Information).

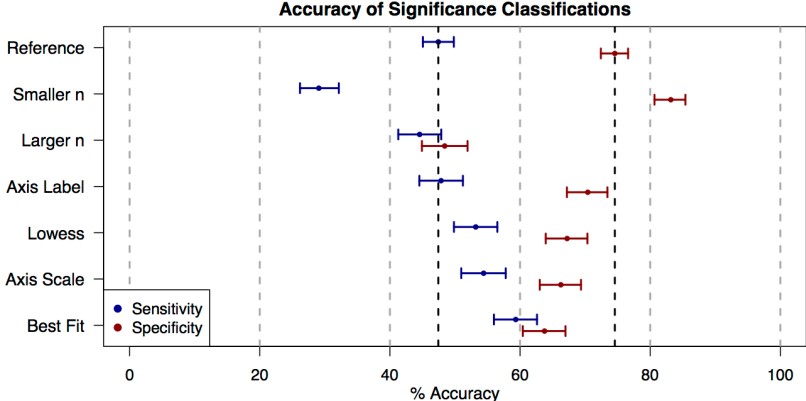

**Figure 2 Accuracy of significance classifications under different conditions.** Point estimates and confidence intervals for classification accuracy for each presentation style (Table 1). Accuracy rates for plots with truly significant underlying relationships (sensitivity) are shown in blue, and accuracy rates for plots with non-significant underlying relationships (specificity) are shown in red.

We found that, overall, subjects tended to be conservative in their classifications of significance. In the reference category (100 data points; Table 1, examples in Fig. 1), students accurately classified graphs of significant relationships as significant only 47.4% (95% CI [45.1%–49.7%]) of the time (i.e., 47.4% sensitivity) and accurately classified graphs of non-significant relationships as non-significant 74.6% (95% CI [72.5%–76.6%]) of the time (i.e., 74.6% specificity) (Fig. 2). Specificity exceeded sensitivity across all of the plot categories presented (Fig. 2).

When comparing the reference plot category of 100 data points to other plot categories (Table 1), sensitivity and specificity were in some cases significantly changed by the number of points displayed in the graph or the style of graph presentation (Fig. 2). Changes to the plots that increased sensitivity correlated with changes that decreased specificity. For example, reducing the number of data points shown ("Smaller *n*" plot category) significantly decreased sensitivity (Odds Ratio (OR) = 0.454, 95% CI [0.385–0.535]) and increased specificity (OR = 1.67, 95% CI [1.39–2.04]). Adding visual aids (best-fit line, lowess curve) significantly improved sensitivity (OR = 1.62 and 1.26 respectively, with 95% CIs [1.38–1.89] and 1.08–1.47), but significantly reduced specificity (OR = 0.600 and 0.699 respectively, with 95% CIs [0.508–0.709] and 0.590–0.829). Changing the scale of the axes also increased sensitivity (OR = 1.32, 95% CI [1.13–1.55]), but decreased specificity (OR = 0.670, 95% CI [0.567–0.792]). Finally, changing the axes label had no significant effect on sensitivity (OR = 1.02, 95% CI [0.871–1.19]) and only a marginally significant effect on specificity (OR = 0.811, 95% CI [0.682–0.965]). Because any gain in either specificity or sensitivity tended to come at the cost of the other, none of these plot categories represented a uniform increase in accuracy across all true significance levels of the data underlying the plots.

The exception to this counter-balancing trend came in "Larger *n*" plots of 200 data points, where students showed a significant drop in specificity (OR = 0.320, 95% CI [0.271–0.377]), and no significant change in sensitivity (OR = 0.891, 95%

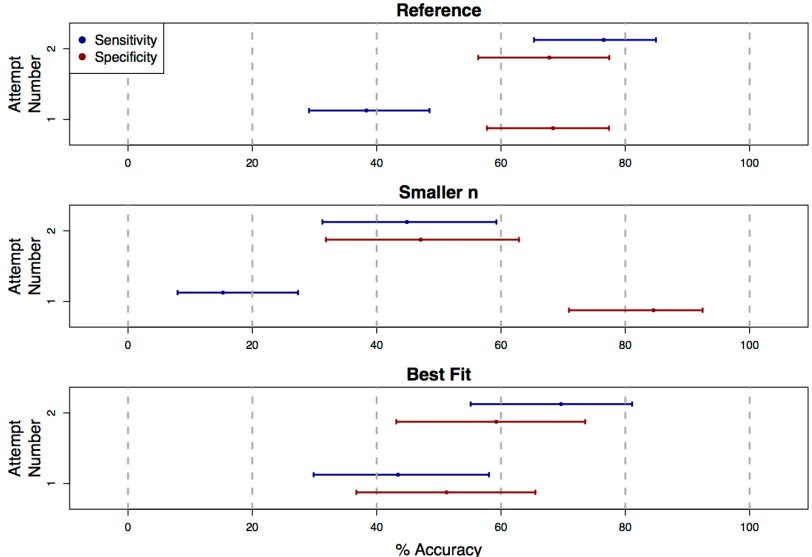

**Figure 3  Classification accuracy on repeat attempts of the survey.** Each plot shows point estimates and confidence intervals for accuracy rates of human visual classifications of statistical significance on the first and second attempt of the survey. For the truly significant underlying *P*-values, users showed a significant increase in accuracy (sensitivity) on the second attempt of the survey for the "Reference," "Smaller *n*," and "Best Fit" presentation styles. For non-significant underlying *P*-values, accuracy (specificity) decreased significantly for the "Smaller *n*" category. Because these accuracy rates were estimated only based on the data from students who submitted more than one response to the survey, the confidence intervals here are wider than those in Fig. 2.

CI [0.763–1.04]). For plots in this category, the probability that users would classify a relationship as significant was fairly similar across truly significant plots and nonsignificant plots. One possible explanation for this is that larger samples require a lower correlation to attain the same significance level. If the correlation becomes imperceptibly small, then the probability that an observer classifies a relationship as significant might be less affected by the true significance level of the plot.

To test if accuracy in visually classifying significance improved with practice, we selected only the students who submitted the quiz multiple times (101 students) and compared accuracy rates between these students' first and second attempts. Of these students, 92% completed their first attempt of the survey, and 99% completed their second attempt of the survey. Because these students self-selected to take the survey twice, they may not form a representative sample of the broader population. However, they may still be representative of motivated students who wish to improve their statistical skills.

We found that, for the "Reference", "Best Fit", and "Smaller *n*" categories, sensitivity improved significantly on the second attempt of the survey (OR = 5.27, 2.98, and 4.51, with 95% CIs [2.69–10.33], [1.28–6.92], and [1.79–11.37]; Fig. 3). For the "Reference" and "Best Fit" categories, the sensitivity improvements were not associated with significant changes in specificity, indicating an improvement in overall accuracy in the visual classification of significance. In the "Smaller *n*" plot category however, the increased sensitivity came at the cost of a significant decrease in specificity (OR = 0.163,

95% CI [0.059–0.447]). For plots in this "Smaller *n*" category, practice did not necessarily improve overall accuracy in visually classifying significance, but rather increased a student's odds of classifying any graph as "significant", regardless of whether the relationship it displayed was truly significant. It is possible that this was due to students over-correcting for their conservatism on their first attempts of the survey. For remaining plot categories ("Larger *n*", "Axis Label", "Lowess", "Axis Scale"), there were no statistically significant changes in sensitivity or specificity between first and second attempts.

## DISCUSSION

Our research focuses on the question of how accurately statistical significance can be visually perceived in scatterplots of raw data. This work is a logical extension of previous studies on the visual perception of correlation in raw data scatterplots (*Cleveland, 1982*; *Meyer & Shinar, 1992*; *Rensink & Baldridge, 2010*), and on the visual perception of plotted confidence intervals in the absence of raw data (*Belia et al., 2005*). The results of this trial are not only relevant towards anyone who wishes to more intuitively understand *P*-values in scientific literature, but also towards designers and observers of scatterplots. Designers of plots should keep in mind that adding trend lines to a plot tends to make viewers more likely to perceive the underlying relationship as significant, regardless of the relationship's actual significance level, so that they can prevent their plots from misleading viewers. Similarly, viewers of scatterplots may want to slightly discount their perception of statistical significance when trend lines are shown.

Our results also suggest that, on average, readers can improve their ability to visually perceive statistical significance through practice. Our intuition is that this improvement is better explained by an improved understanding of what effect sizes constitute significant relationships, rather than an improved ability to visually distinguish these effect sizes. It would follow that the apparent baseline poor accuracy in visually detecting significance is largely due to a false intuition for what constitutes significant relationships. A broad movement towards practicing the task of visually classifying significance could improve this intuition, and better the efficiency and clarity of communication in science. To help readers train their sense for *P*-values, we've created an interactive online application where users can explore the connection between the significance level of a bi-variate relationship and how the data for that relationship appears in a scatterplot (http://glimmer.rstudio. com/afisher/EDA/). Users can see the visual effect of changing sample size while holding the *P*-value constant. They can also add lowess curves and best-fit lines to the scatterplot.

This work is also relevant to debate over the misuse of EDA. It has been argued that when EDA and formal hypothesis testing are applied to the same dataset, the "data snooping" committed through EDA process can increase the Type I error rates of the formal hypothesis tests (*Berk, Brown & Zhao, 2010*). However, the apparently low sensitivity with which humans can detect statistically significant relationships in scatterplots implies that both the costs of EDA misuse, as well as the benefits of responsibly conducted EDA, may be smaller than expected.

Data analysis involves the application of statistical methods. Our study highlights that even when the theoretical properties of a statistic are well understood, the actual behavior in the hands of data analysts may not be known. Our study highlights the need for placing the practice of data analysis on a firm scientific footing through experimentation. We call this idea of evidence based data analysis, as it closely parallels the idea of evidence based medicine, the term for scientifically studying the impact of clinical practice. Evidence based data analysis studies the practical efficacy of a broad range of statistical methods when used, sometimes imperfectly, by analysts with different levels of statistical training. Further research in evidence based data analysis may be one way to reduce the well-documented problems with reproducibility and replicability of complicated data analyses.

### Funding

This research was partially supported by the National Institute of Environmental Health Sciences (grant number T32ES012871). The funders had no role in study design, data collection and analysis, decision to publish, or preparation of the manuscript.

### Grant Disclosures

The following grant information was disclosed by the authors:
National Institute of Environmental Health Sciences: T32ES012871.

### Competing Interests

The authors declare there are no competing interests.

### Author Contributions

- Aaron Fisher conceived and designed the experiments, performed the experiments, analyzed the data, contributed reagents/materials/analysis tools, wrote the paper, prepared figures and/or tables, reviewed drafts of the paper.
- G. Brooke Anderson and Jeff Leek conceived and designed the experiments, performed the experiments, contributed reagents/materials/analysis tools, wrote the paper, reviewed drafts of the paper.
- Roger Peng conceived and designed the study and helped write the paper.

### Human Ethics

The following information was supplied relating to ethical approvals (i.e., approving body and any reference numbers):

Johns Hopkins Bloomberg School of Public Health IRB Approval number: IRB00005072, 45 CFR 46.101(b)(4).

### Data Deposition

The following information was supplied regarding the deposition of related data:
https://github.com/aaronjfisher/visual_pvalue/tree/master.

## Supplemental Information

Supplemental information for this article can be found online at http://dx.doi.org/10.7717/peerj.589#supplemental-information.

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
