# Peer review of "A randomized trial in a massive online open course shows people don’t know what a statistically significant relationship looks like, but they can learn"

_PeerJ, doi:10.7717/peerj.589_

## Round 0.1 · original submission · Minor Revisions

You can see that the reviewers think this is a really exciting study and are eager to see it published. They have some minor revisions related to the presentation of the results and the statistics (e.g., correlation vs p-values) used, and a more thorough description of the study design. They both performed a really thorough review of the paper and supplement and made some salient observations about the presentation of the material that I think you can easily address in a revision.

·

Basic reporting

Figure 1: Plotting the sensitivity and specificity on the same plot (using different colors as already done) would allow for easier visual comparison of the two confidence intervals. In addition, this would allow for using the entire width of the paper for the x-axis, and hence make it easier to figure out the span of the confidence intervals. The figure would also be improved by adding vertical dotted lines at 0, 20, 40, 60, 80, 100 in gray, once again to make it easier to identify the endpoints of the intervals. I am also unclear as to why the ``Reference" isn't the first entry on the plot. Unless there is a specific reason, I would recommend moving it to the top so that it's clear that the comparisons discussed later in the paper refer to comparing each of the categories to the reference level.

Figure 2: It is not clear why the dots are shown on these plots. They appear to be jittered versions of a points that should be at (1,0), (1,1), (2,0), and (2,1) for attempt 1 / incorrect answer, attempt 1 / correct answer, attempt 2 / incorrect answer, and attempt 2 / correct answer, respectively. The point estimate for each attempt summarizes these data and the y-values of the points themselves (after jitter) is misleading. I understand the purpose of jittering (otherwise there would only be 4 points on the plot at the coordinates listed above) however perhaps the only thing that's relevant about these points is how many of them there are at each coordinate. This could be communicated much more efficiently by listing these counts, or plotting a single character at the coordinates listed above where the size of the plotting character indicates the number of observations that fall into that bin. I strongly suggest revisiting this plot and making revisions as suggested here, or alternative revisions that meet the same goals.

Stylistic comment: I recommend using the leading 0s in decimal values between 0 and 1. For example ``OR=.670" should be replaced with ``OR = 0.670". The latter is much easier on the eye when reading the text.

Specific comments on the text:
1. p.1: “consuming data analytic reports” = “consuming data analysis reports”
2. p.1: The first paragraph lists “A lack of sufficient training in statistics and data analysis has been responsible for the retraction of high-profile papers, the cancellation of clinical trials, and mistakes in papers used to justify major economic policy initiatives.” I would recommend adding citations to each of these examples, perhaps from media coverage referring to each of the items. I’m guessing, for example, that the last item refers to the recent findings relating to the austerity movement.
3. p.1: “The real effect of a statistical procedure depends to a large extent on psychology and cognitive function.” = “The real effect of a statistical procedure depends, to a large extent, on psychology and cognitive function.”
4. p.2: “was changed by the number of data points in the plot or way the plot was presented” = “was changed by the number of data points in the plot or the way the plot was presented”
5. p.2: “in the examples plots shown in the top panel of Figure 1” = “in the example plots shown in the top panel of Figure 1”
6. p.3: “In the reference category (100 data; Table 1, examples in Figure 1 top panels)” = “In the reference category (100 data points; Table 1, examples in Figure 1 top panels)”
7. p.3: “When comparing the reference plot category of 100 datapoints” = “When com- paring the reference plot category of 100 data points”. Table 1 uses two words (“data points”) and the text uses both versions (“data points” and “datapoints”). The correct spelling should be two words, or at a minimum there should be consistency in the text. This change should be implemented in multiple spots in the text.
8. p.3: “For example, reducing the number of datapoints” = “For example, reducing the number of data points”
9. p.3: “(“Lower n” plot category)” ?= “(“Smaller n” plot category)” → There is no “Lower n” plot category in Figure 1. This change should be implemented in multiple spots in the text. Alternatively the category shown in the plot can be updated to match the text.
10. p.5: “For plot in this “Lower n” category” = “For plots in this “Lower n” category” + whatever change is adopted for “Lower/Smaller n” (discussed above in item 9).
11. p.5: “For remaining plot categories (“Higher n”,” = “For remaining plot categories (“Larger n”,” This is similar to the comment in item 9, there is no “Higher n” category in Figure 1, so either change the text or the figure to match Higher/Larger n.
12. p.5: “in raw data scatterplots [5, 14, 15],and on the visual” = “in raw data scatterplots [5, 14, 15], and on the visual” (add space between comma and the next word)
13. p.5: “adding trend lines in a plot” = “adding trend lines to a plot”

Experimental design

I would recommend describing how these students were selected. Was this a voluntary sample, and if so, how were the students recruited? Or were the students selected randomly from the entire course roster, and if so, what was the response rate?

A general description of the course as well as how far into the course the experiment was conducted would also be useful for gauging the statistical background of the students prior to being subjected to the experiment. The authors state that ``All students in the class had been exposed to lecture material explaining P-values and null hypothesis significance testing prior to the study." Considering that all MOOC students do not necessarily watch all lecture videos, I am not sure if this is an accurate statement. Revising this to state that such lecture material had been made available for the students would be an improvement, unless it has been ensured that all students who participated in this experiment actually did watch the relevant videos. If so, it should be stated that a mechanism was in place to ensure this.

Validity of the findings

The authors in their conclusion state that ``Our study highlights that even when the theoretical properties of a statistic are well understood, the actual behavior in the hands of data analysts may not be known." Is the statistic referred to here the p-value? If so, has there been a pre-test done to ensure that the subjects in this experiment well understand the theoretical properties of this statistic? If not, this conclusion seems to be a stretch based on this study.

Lastly, while I agree with the sentiment that ``Further research in evidence based data analysis may be one way to reduce the well-documented problems with reproducibility and replicability of complicated data analyses.", further justification is needed to support the claim that findings of this particular study can help reduce problems with reproducibility and replicability.

·

Basic reporting

Main text:
* It would be nice to have citations for "A lack of sufficient training in statistics and data analysis has been responsible for the retraction of high-profile papers, the cancellation of clinical trials, and mistakes in papers used to justify major economic policy initiatives."
* There is an Extra comma in "(OR=5.27, 2.98, and , 4.51, with"
* Nice to have leading zeros in places like "OR=.163"
* Extra hyphen in reference 5


In supplement:
* "should be streamlines" -> "should be streamlined"
* Put periods and commas after references
* Some of the references are incomplete / missing capitalization.
* add "as of 2013" to the 5 million people number?
* it says in the code that you are planning to add the knitr output to the
supplement. That would be a great idea (and I love that you published the
code).

Experimental design

* I'm not crazy about phrasing everything in terms of p-values here. It's not
clear to me what influences accuracy more - correlation or p-value (I'd guess
the first). For example, imagine presenting 5 points that are highly
correlated but not significantly so. In any case, I'd be a bit more precise
about the claim in the abstract - you aren't quite testing the ability to
distinguish between statistically significant and not, you are testing the
ability to discern between a mildly significant correlation (p = 0.025, r2 ~
0.05) and a non-significant correlation. Taken a bit further, one might frame
this as a more direct criticism of p-values (perhaps the moral is that a p-value
of 0.025 with only 100 data points is by itself rather meaningless).

* It would be really interesting to see the plot of sensitivity and specificity
at various correlations and sample sizes. Is there a phase transition? Does
p-value or correlation coefficient influence accuracy more? I realize you
haven't collected this, but I'd love to see it in the future.

* How was the survey presented to students? Were they all equally encouraged to
follow-up and take it twice? The response rate was quite low there (only 5%
of people taking it once). Assuming there was substantial flexibility in
which students took it multiple times, what do you think of the possibility
of selection bias? I.e., did only students who were good at the game take the
second round? Did the students see their results?

* I find the fact surprising that there was only 10-15% variation attributable
to individual factors. Do you know more about the students, and if so, did
anything correlate with estimated baseline ability?

Validity of the findings

The statistical methods appear to be sound; the mixed effect model used is appropriate.

Additional comments

This is a very cool, fun paper, and I love the idea of evidence-based data analysis. There are many fun directions that one could follow-up.

---

## Round 0.2 · accepted · Accept

Thank you for addressing all of the reviewers' concerns thoroughly. I am very excited about this research and I am eager to see it published!